# Medulloblastoma: Molecular Targets and Innovative Theranostic Approaches

**DOI:** 10.3390/pharmaceutics17060736

**Published:** 2025-06-04

**Authors:** Alice Foti, Fabio Allia, Marilena Briglia, Roberta Malaguarnera, Gianpiero Tamburrini, Francesco Cecconi, Vittoria Pagliarini, Francesca Nazio, Adriana Carol Eleonora Graziano

**Affiliations:** 1Department of Medicine and Surgery, University of Enna “Kore”, 94100 Enna, Italy; alice.foti@unikore.it (A.F.); fabio.allia@unikore.it (F.A.); roberta.malaguarnera@unikore.it (R.M.); 2Department of Neuroscience, Sensory Organs, Chest—Pediatric Neurosurgery, Fondazione Policlinico Universitario A. Gemelli IRCCS, 00168 Rome, Italy; gianpiero.tamburrini@policlinicogemelli.it; 3Fondazione Policlinico Universitario A. Gemelli IRCCS, 00168 Rome, Italy; francesco.cecconi@unicatt.it; 4Department of Basic Biotechnological Sciences, Intensivological and Perioperative Clinics, Università Cattolica del Sacro Cuore, 00168 Rome, Italy; 5Department of Neuroscience, Section of Human Anatomy, Università Cattolica del Sacro Cuore, 00168 Rome, Italy; vittoria.pagliarini@unicatt.it; 6GSTEP-Organoids Research Core Facility, Fondazione Policlinico Universitario A. Gemelli IRCCS, 00168 Rome, Italy; 7Department of Biology, University of Rome Tor Vergata, 00133 Rome, Italy; francesca.nazio@uniroma2.it

**Keywords:** nanomedicine, brain tumors, blood–brain barrier, rare tumors

## Abstract

**Background/Objectives:** Medulloblastoma is a rare tumor that represents almost two-thirds of all embryonal pediatric brain tumor cases. Current treatments, including surgery, radiation, and chemotherapy, are often associated with adverse effects, such as toxicity, resistance, and lack of specificity. According to multiple bulk and single-cell omics-based approaches, it is now clear that each molecular subgroup of medulloblastoma possesses intrinsic genetic and molecular features that could drive the definition of distinct therapeutic targets, and of markers that have the potential to improve diagnosis. Nanomedicine offers a promising approach to overcome these challenges through precision-targeted therapies and theranostic platforms that merge diagnosis and treatment. This review explores the role of nanomedicine in medulloblastoma. Here, possible theranostic nanoplatforms combining targeted drug delivery and simultaneous imaging are reviewed, highlighting their potential as tools for personalized medicine. **Methods:** We performed a chronological analysis of the literature by using the major web-based research platforms, focusing on molecular targets, and the potential application of nanomedicine to overcome conventional treatment limitations. **Results:** Advances in nanoparticle-based drug delivery systems enable selective targeting of key molecular pathways, improving therapeutic efficacy while minimizing off-target effects. Additionally, nanotechnology-based imaging agents, including MRI contrast agents and fluorescent probes, improve diagnostic accuracy and treatment monitoring. Despite these advantages, some significant challenges remain, including overcoming the blood–brain barrier, ensuring biocompatibility, and addressing regulatory pathways for clinical translation. **Conclusions:** In conclusion, we sought to identify the current knowledge on the topic and hope to inspire future research to obtain new nanoplatforms for personalized medicine.

## 1. Introduction

One of the most prevalent pediatric brain tumors in the posterior cranial fossa is medulloblastoma (MB), accounting for 20% of all pediatric brain tumors (WHO grade IV) [1]. The majority of MB cases occur in children between 3 and 7 years. Although the 5-year overall survival rate for MB is around 75%, there are still major concerns about long-term therapy-related complications [2]. In modern classification, MB is recognized as a heterogeneous tumor with multiple subtypes, all sharing a common primitive embryonal phenotype, consisting of malignant cells primarily characterized by neuronal antigen expression. From a biological point of view, the development and heterogeneity of MB involve the overproliferation of progenitor neural cells of the rhombic lip and a lineage shift towards cells with a malignant phenotype (Figure 1).

Genome-wide high-throughput analyses enabled MB classification into four distinct molecular subgroups: Wingless (WNT), Sonic Hedgehog (SHH), Group 3 (G3), and Group 4 (G4) [3,4]. Each group possesses unique transcriptional and epigenetic profiles, as well as genetic mutations and clinical outcomes. WNT MB is caused by mutations which constitutively activate the WNT signaling pathway, representing 9–10% of cases and which among all subgroups is the one with the most favorable prognosis. The SHH subgroup accounts for 28–30% of cases and, like the previous subgroup, is caused by constitutive activation of a signaling pathway (i.e., the SHH signaling pathway). The G3 and G4 MB subgroups represent the remaining 19–25% and 35–43% of cases, respectively [5].

For patients with MB, the evaluation of a tumor’s anatomical spread at the time of diagnosis is crucial [6]. Nowadays, current treatments rely on surgical resection along with radiation and chemotherapy, with five-year survival rates of between 50% to 90% [7]. Traditional treatments, however, cause short-term and long-term adverse effects, altering patients’ quality of life [8]. Even with progress in surgical methods, radiotherapy, and the identification of novel chemotherapy drugs, the outlook for individuals with intracranial tumors, especially those situated in crucial brain areas or with high malignancy, remains bleak. Tumors in these regions pose distinct challenges because of their closeness to brain areas with essential biological and physiological functions, making surgical procedures more complex. Moreover, conventional treatments are often associated with adverse effects, such as toxicity and lack of specificity towards the tumor district. Consequently, there is increasing interest in targeted drug therapy, which has the potential to specifically attack tumor cells while minimizing harm to nearby healthy tissues. Based on these assumptions, nanomedicine approaches have been investigated to overcome these limitations and obtain an efficient and personalized medicine. Moreover, nanomedicine could be useful to create theranostic platforms that merge diagnosis and treatment.

This review aims to highlight the characteristics of the MB molecular subgroups, linking them to the putative role of nanomedical systems in their treatment and monitoring. Therefore, in this review, we first introduce readers to the topic of MB by focusing on the histological and molecular features of the MB subgroups. Then, we critically analyze the major applications of nanomedicine in the context of brain tumors, with a critical evaluation of the distinct advantages for biomedical applications linked to targeting strategies and/or to the physicochemical properties of nanoparticles (NPs). These findings support the last part of the work, in which we chronologically analyze the scientific literature by reviewing the in vitro, in vivo and clinical evidence on nanomedicinal approaches for MB. A particular effort also was made to organize and critically revise all the available data on the use of nanoparticles (NPs) for theranostic approaches in MB, combining targeted drug delivery and simultaneous imaging. Since nanomedicine approaches are developing rapidly, a chronological overview of research on MB is presented here.

## 2. Medulloblastoma: The Molecular Subgroups

Despite the common primitive embryonal phenotype, genome-wide high-throughput analyses have enabled MB classification into four distinct molecular subgroups, as shown in Figure 2 and described below.

### 2.1. Wingless

The vast majority of WNT MBs examined so far exhibit classic histological features. Compared to the other subtypes, WNT MB shows an altered vascular environment, with downregulation of genes normally expressed in the central nervous system (CNS) endothelium and upregulation of genes physiologically expressed in the peripheral endothelium [9]. As a consequence, endothelial tight junctions are also compromised, showing regions with reduced junctional density. These features result in altered blood–brain barrier (BBB) functionality [10].

Somatic catenin beta 1 (CTNNB1) mutations are the most common alterations in the WNT subgroup (almost 90%), followed by the loss of one copy of chromosome 6 (monosomy 6) (almost 86%) [11,12]. CTNNB1, the gene encoding for β-catenin mapped on the short arm of chromosome three (3p22.1), presents mutations located at exon 3, particularly at the phosphorylation site of β-catenin. Therefore, β-catenin phosphorylation is inhibited, leading to its cytoplasmic accumulation and translocation to the nucleus, where it activates genes involved in cell proliferation [13]. WNT MB lacking CTNNB1 mutations are related to a germline pathogenic variant of adenomatous polyposis coli (APC), a repressor of WNT signaling, which causes a genetic disorder called Turcot syndrome. APC is part of a complex that is responsible for the phosphorylation-dependent ubiquitylation and degradation of β-catenin. Therefore, APC loss-of-function leads to constitutively activated WNT signaling [11] (Figure 2).

Given its favorable prognosis and low-risk biological profile, recent clinical trials are exploring treatment de-escalation strategies doses to minimize off-target effects. For instance, the clinical trials NCT02066220, NCT01878617, and NCT02724579 are investigating the use of reduced-dose craniospinal radiation and chemotherapy [14].

### 2.2. Sonic Hedgehog

The Hedgehog signaling pathway is involved in the development process of the CNS, particularly in the regulation of embryonic cell differentiation, granule neuron progenitors (GNP) proliferation, and maintenance of the self-renewal capacity of neural stem cells [15].

In particular, SHH MB arises from cerebellar GNP cells [16], which rely on SHH pathway signaling for their expansion during the perinatal period [16,17,18]. Various mutations can lead to activation of the Hedgehog pathway in a ligand-independent manner, including (i) loss of function of the negative regulators Patched1 (PTCH1) and suppressor of fused homologue (SUFU); (ii) gain of function of the positive regulator Smoothened (Smo), (Figure 2). Moreover, mutations of the negative regulators are also correlated to Gorlin syndrome, which causes MB predisposition. The SHH subgroup is also correlated with the overexpression of the epidermal growth factor receptor (EGFR) [19].

SHH MB, unlike WNT MB, is molecularly, histologically, and clinically heterogeneous, with multiple molecular subtypes that influence prognosis, treatment response, and tumor behavior [5]. According to TP53 mutation status, SHH-MB can be histologically divided into desmoplastic/nodular (DN) and the large cell/anaplastic (LCA) variants. The first, with a more favorable prognosis, linked to the TP53 wild type; the second, characterized by TP53 mutations with a more aggressive profile [20].

### 2.3. Group 3

G3 MB is the most aggressive and high-risk subtype, with the worst prognosis. It originates from a rhombic-lip-derived lineage, with a particular expression profile of unipolar brush cells (Figure 2) [21]. Unlike WNT and SHH MBs, G3 has a low frequency of nucleotide variants and germline mutations. The most common molecular alterations involve the amplification and overexpression of the MYC proto-oncogene. The MYC proto-oncogene plays a crucial role in driving cell cycle progression, particularly during the G1 to S phase transition, by regulating the activity of cyclin-dependent kinases (CDKs) [22]. In normal cells, cyclin E/CDK2 activity is tightly regulated by MYC to prevent excessive cell division. However, when MYC is overexpressed, this activity becomes deregulated, leading to uncontrolled cell division [23]. G3 MBs frequently exhibit LCA histology; however, a subset exhibits classic histology [24].

### 2.4. Group 4

G4 MB is the least understood among all MB subtypes. Like the G3 subtype, G4 is not characterized by alteration of a specific molecular pathway; moreover, they share the same histologic features, either classic or LCA histology, although LCA histology is much less common in this group [25]. Moreover, it has been reported that G4 arises from unipolar brush cells, like G3 MB (Figure 2) [26]. Currently, the driver oncogenes and/or tumor suppressor genes linked to this alteration remain unidentified, since there is not a single gene mutated in >10% of cases [27]. The most frequent alteration (about 17% of cases) can be found in an enhanced hijacking mechanism, which leads to the overexpression of the PR/SET domain 6 protein (PRDM6) [26]. PRDM6 is a transcriptional repressor with chromatin-modifying potential activity as histone methyltransferase.

## 3. Nanomedicine for Brain Tumors

The term nanomedicine refers to the application of nanotechnology to medicine. Nanomedicine has revolutionized the approach to treat various diseases, particularly, oncological and neurodegenerative conditions [28,29].

Nanoparticles (NPs) are submicroscopic particles measuring between 1 and 1000 nanometers that have been demonstrated to be promising drug delivery vehicles [30]. First of all, their small dimensions allow them to easily infiltrate the tumor tissue and concentrate at the tumor site, enhancing chemotherapy’s effectiveness [31]. Additionally, the surface properties of nanoparticles can be tailored to optimize their interaction with biological systems [32,33]. This includes modifying the surface charge [34], hydrophobicity [35], or adding specific ligands to enhance targeting capabilities and reduce non-specific interactions [36,37]. Therefore, their unique characteristics have made them revolutionary tools in drug delivery and therapy. Drug delivery encompasses the methods and systems designed to transport medicinal agents to specific body locations in a regulated manner [38]. At the same time, NPs can be used as therapeutic agents themselves to combat diseases by interacting with biological systems to achieve a desired outcome [39].

Conventional cancer treatments like chemotherapy often cause substantial side-effects due to their non-specific nature, affecting both cancerous and healthy cells. In contrast, NPs can be modified with ligands or antibodies that bind to specific cell surface receptors, facilitating targeted drug delivery to cancer cells. This approach also shields active ingredients from degradation and immune system clearance, resulting in extended drug circulation times and improved efficacy. Depending on their type, origin, and source, NPs allow different targeting strategies by tailoring their characteristics according to the biological needs.

A concise and critical overview of the most commonly used nanomedical models for brain tumors is essential in order to guide readers toward the nanomedicine-based strategies that have been reported for MB.

### 3.1. Nanomedicine for Brain Tumors Targeting Strategies

One of the major challenges in brain tumor treatment is overcoming the BBB, which represents a semi-dynamic permeable membrane to ensure homeostasis of the CNS [40], controlling the outflow and inflow of substances. Anatomically, the BBB is formed by endothelial cells of the brain microvessels, astrocytes, pericytes, tight junctions, and the basement membrane. The latter controls the passage of substances in and out [41]. Moreover, the surface of endothelial cells is rich in transporters, such as glucose transporters (GLUT), excitatory amino acid transporters (EAAT1–3), and the transferrin receptor (TfR), which can be used to specifically target the BBB [42]. Molecules with molecular weight <500 Da that are lipophilic and charged are able to cross the BBB by passive diffusion [43]; however, most drugs do not fit these conditions. Therefore, a strategy to overcome this problem is needed. It is important to take into consideration that brain tumors are associated with BBB dysfunction, with altered characteristics with respect to integrity, vascularization, and efflux pump expression, representing a challenge when it comes to achieving a successful therapeutic strategy.

In MB, BBB shows low capillary permeability and blood flow even where capillary density is not uniform [44]. The only subgroup which shows an impaired BBB is the WNT, which is also the subgroup with the best prognosis [10]. This characteristic makes it difficult to achieve an effective therapy. The standard treatment for MB is based on surgery, radiation therapy, and chemotherapy [45]. Unfortunately, chemotherapeutics have difficulties in overcoming the BBB; therefore, alternative strategies are needed.

Several strategies, divided into invasive and non-invasive, have been used to determine a drug’s concentration inside the BBB to ensure the best approach to reduce damage in the CNS and implement the pharmacokinetic functions of the drug [46]. In this context, NPs offer the possibility to target the brain by passive or active mechanisms (Figure 3).

#### 3.1.1. Passive Targeting

In order to achieve effective drug delivery, it is important to understand the pathophysiological features of the tumor environment. In solid tumors, the new blood vessels are different from those of normal tissues, which are dense. Hence, there are gaps between endothelial cells. The enhanced permeability and retention (EPR) effect is a consequence of the different vascularization, which causes the accumulation of molecules of certain sizes in tumor tissue [47,48]. Therefore, passive targeting exploits the EPR effect to facilitate the delivery, accumulation, and retention of molecules/nanoparticles in the tumor site [49]. This phenomenon is influenced by NPs size, shape, and surface properties, which need to be finely tuned to obtain the best performance [50]. Although this strategy appears promising, actual clinical application has not yet been consistently achieved [51].

#### 3.1.2. Active Targeting

In contrast to passive targeting, active targeting involves surface modification of NPs with specific ligands, antibodies, or peptides that recognize and bind to overexpressed receptors on brain tumor cells or to BBB receptors. This allows not only their efficient localization in the tumor tissue, but also their cellular uptake if internalizing receptors are targeted [52]. This strategy improves drug delivery efficiency, minimizes off-target effects, and increases therapeutic efficacy by ensuring that nanoparticles accumulate preferentially in tumor tissue rather than in healthy brain regions.

EGFR is a tyrosine kinase receptor which is overexpressed in human brain tumors, including MB; therefore, it can be used as a target to functionalize NPs [53]. Currently, there is no EGFR-decorated nanoformulation that is specifically designed for MB; however, some studies have been performed on glioblastoma (GBM). Kaluzova et al. [54] studied the effects of cetuximab iron oxide nanoparticles (IONPs) against EGFR-expressing glioma stem cells. These IONPs have been reported to be internalized and effective in inducing cells to apoptose by binding to EGFR, inhibiting its signaling pathway. EGFR targeting has also been used in combination with low-density lipoprotein receptor-relative protein-1 (LRP1), which is usually expressed in BBB endothelial cells [55]. This strategy, by Liu et al., allowed both BBB penetration and targeting of tumor cells [55].

Another receptor used to target brain tumors is the transferrin receptor (TfR), which has the role of transporting iron into cells; therefore, it is highly expressed in various body areas, including both normal brain cells and brain cancers [56]. Again, many studies have been performed on GBM, finding that TfR is a good target for crossing the BBB and delivering drugs [57].

Lipoproteins of both low (LDL) and high (HDL) density can also be used for targeted therapy. For instance, Nikanjam M. et al. [58] developed a nano-LDL particle to target GBM cells. On the other hand, Bell et al. [59] developed HDL NPs to target SHH MB and deprive cells of natural HDL.

### 3.2. Nanoparticle Classification

Nanoparticles (NPs) can be systematically classified according to their chemical composition. The specific physicochemical properties translate into distinct advantages for biomedical applications. According to the literature, organic NPs are mainly used for drug delivery, while inorganic NPs are used for imaging and for radio/photothermal therapy. The encapsulation of inorganic NPs into organic nano-compartments or mixed lipid-polymer carriers results in hybrid systems with enhanced potential applications in drug delivery, bioimaging, and theranostics [60]. Biomimetic NPs are mainly applied to targeted drug delivery, enabling reduced drug dose to achieve therapeutic effects. Figure 4 summarizes the main types of NPs studied for their application in the context of brain tumors.

#### 3.2.1. Polymeric Nanoparticles

Polymers, being macromolecular structures, can exhibit different properties depending on their composition. They are formed by covalent bonding of monomers with various structures and functionality, allowing the creation of linear or branched chains with tailored characteristics. The customization potential of polymers enables them to be used for multiple applications. Therapeutic molecules can be encapsulated, adsorbed or conjugated with polymeric NPs. In this way, drugs are protected from enzymatic degradation and rapid elimination, contributing to good pharmacokinetic and pharmacodynamic properties, and offering controlled release of the drug through external stimuli such as pH and temperature. The polymer shell can be modified with various molecules for cell targeting purposes. Diagnostic tools can also be obtained by tagging polymers with fluorescent dyes [61]. One example of a widely used polymeric NP is that made of polylactic-co-glycolic acid (PLGA), which has been determined to be an efficient drug carrier in a glioma mouse model [62].

Progress in the field of polymeric NPs is ongoing, reflecting the multiple possibilities to design and combine molecules to obtain the desired characteristics. For example, Fukui et al. used chitosan-pegylated NPs conjugated with folic acid to deliver siRNA in a mouse glioblastoma model [63]. Moreover, polymeric micelles made of cationic polymers, such as polyethyleneimine (PEI), combined with hydrophobic polymers are used to deliver negatively charged nucleic acids and hydrophobic cancer drugs. Cheng et al. showed that PEI and polycaprolactone (PCL) micelles functionalized with folate are able to deliver BCL-2 siRNA and doxorubicin (DOX) in C6 glioma tumors in rats, increasing apoptosis of tumoral cells [64].

Recent advances in polymer science have led to the development of architecturally complex molecular systems for brain tumors. Among them, dendrimers are polymeric sphere-shaped NPs, with hyperbranched and symmetrical structures. Their structure can be divided into three components: a central core, branches, and terminal groups. Thanks to their numerous hydrophobic compartments, they are able to encapsulate different drugs, allowing their controlled release at the desired target. There are different types of dendrimers, including polyamidoamine (PAMAM), polypropylenimine (PPI), etc., which can either be used for passive [65,66] or active targeting strategies [67]. PAMAM dendrimers conjugated with folic acid were used to release DOX in a C6 glioma xenograft rat model [68]. Globally, these data suggest that the functionality of polymeric NPs can be modulated and targeted by working on the NP architecture. Especially in cases of complex structures, it should be noted that the multistep processes required for their synthesis often result in batch-to-batch variability—an important issue for clinical application. As PAMAM dendrimers exhibit toxicity due to their high cationic charge density, peptide-based asymmetric dendrimers have been developed to minimize their toxicity profile, maintaining the possibility of efficient cell transfection, especially of anionic genes (DNA or RNA) [69].

In this context, other natural biopolymers, such as polysaccharides (chitosan, fucoidan, etc.), have been widely used to develop biodegradable and biocompatible NPs for the encapsulation and delivery of bioactive substances. These systems have the advantage that, in some cases, the biopolymers exert their own biological activity [70].

#### 3.2.2. Lipid-Based Nanoparticles

Lipid NPs can be divided into two categories: micelles and liposomes. Micelles are spheres made of amphiphilic molecules with a hydrophilic surface and a hydrophobic core that can encapsulate drugs [71]. Liposomes, on the other hand, have a lipid bilayer and are able to transport both hydrophilic (inside the core) and hydrophobic (within the lipid bilayer) drugs [72].

Liposomes can deliver drugs either via passive or active targeting. Free drugs can be accidentally delivered to healthy tissue; however, by increasing the size by encapsulating them into liposomes, they can enter and accumulate at the tumor site due to the EPR effect [73]. The main problem in the use of liposomes is that they are subject to removal mediated by macrophages in a way that depends on their size, surface charge, and cholesterol content [74]. One of the strategies to avoid liposome clearance is to link them with polyethylene glycol (PEG), preventing an immune response towards NPs [75]. Active targeting can be achieved by functionalizing liposomes with antibodies [76], peptides [77], or aptamers [78].

#### 3.2.3. Inorganic Nanoparticles

Inorganic NPs have distinct characteristics, such as a large surface-area-to-volume ratio, small size, and high drug loading capacity [79]. Compared to other nanoparticles, they have intrinsic physicochemical properties (magnetic, thermal, optical, and catalytic); however, problems with biodistribution can occur [80]. In the context of brain tumors, NPs based on gold (AuNPs), silver (AgNPs), and iron oxide (IONPs), and quantum dots (QDots), have been increasingly explored.

AuNPs can be used both for diagnostic and therapeutic applications because of their small size, which allows crossing of the BBB by passive diffusion or carrier/receptor-mediated endocytosis [81]. Moreover, their tunable shape (e.g., nanorods, nanodiamonds, etc.) allows them to absorb light in the near-infrared (NIR) range and to be used for photothermal therapy [82]. Jensen et al. [83] used AuNPs to deliver siRNA in xenograft models of GBM to target the oncoprotein Bcl2Like12 (Bcl2L12), a p53 inhibitor, achieving cell apoptosis without side-effects. AuNPs can also be used as imaging tools for brain tumor diagnosis [84].

AgNPs are known mostly for their antibacterial and anticancer effects caused by the release of Ag^+^ ions, which generate reactive oxygen species (ROS), and cause cellular oxidative damage and cell death [85]. Akyuva et al., for instance, studied the combination of AgNPs and cisplatin, which caused oxidant and apoptotic actions in GBM cells. Moreover, Salazar-García et al. showed a toxic effect of AgNPs on rat glioma cells [86].

IONPs are a class of NPs which find application in cancer therapy for drug delivery and as magnetic resonance imaging (MRI) contrast agents [87]. Moreover, IONPs have unique advantages in specific targeting using a magnetic field. The same feature can be exploited by using an alternating magnetic field to kill cells through hyperthermia [88]. For example, Hadjipanayis C. et al. [89] studied superparamagnetic IONPs coupled with an anti-EGFR antibody to target GBM and use them for MRI.

Quantum dots (QDots) [90] are semiconductor core-shell structures with unique electronic and optical properties. Unfortunately, their clinical use is still limited due to their non-specific cytotoxicity [91]. An attempt to exploit the properties of QDots was made by Wang X. et al., who developed polydopamine (PDA)-coated QDots to deliver and release the drug temozolomide in an acidic tumor microenvironment with simultaneous real-time visualization via fluorescence imaging [92].

Inorganic nanoparticles offer significant potential application for active targeting and drug delivery as they can be functionalized with targeting ligands, or used in bioimaging, enabling real-time tumor visualization and monitoring as MRI contrast agents, and in therapy, based on their physical properties. Despite these advances, non-biodegradable inorganic NPs can accumulate, and from a clinical perspective, their long-term toxicity is unpredictable, limiting their application.

#### 3.2.4. Biomimetic NPs

Biomimetic NPs are a new class of nanomaterials, whose structure is incorporated with naturally derived components—such as cell membranes, lipoproteins, or serum proteins. By imitating the surface characteristics and biological behavior of endogenous cells or molecules, these systems achieve superior biocompatibility and higher retention times. Among these, serum albumin has been investigated for nanoparticle fabrication. Albumin is non-toxic and highly stable in body fluids, has high drug-binding capacity, and is biodegradable; therefore, it offers several advantages in the enhancement of drugs’ half-lives. Moreover, it is already approved by the U.S. Food and Drug Administration for pharmaceutical applications [93]. For example, paclitaxel-loaded human serum albumin (HSA) NPs modified with substance P peptide as the targeting ligand were analyzed for their antitumor effect on GBM [94]. Another example of biomimetic NPs is of natural high-density lipoproteins (HDLs), which are the smallest of the plasma lipoproteins and can be recognized by diverse receptors [95]. HDL was used as a nanoplatform for the co-delivery of a fluorescent dye and a small interfering RNA (siRNA) for Hypoxia-inducible factor 1-alpha (HIF-1α) for enhanced photo-gene therapy towards glioma. This study showed how HDL nanoplatforms were able to efficiently cross the BBB and inhibit tumor growth without side-effects [96].

## 4. Nanomedicine and Medulloblastoma: A Chronological Overview

According to the literature reviewed, the first study on NPs and MB was performed in 2006 by Meng W. et al. [97]. In the last two decades, nanomedicine has emerged as a potential approach in different aspects of MB management. Figure 5 summarizes the rational design and functionalization of NPs for MB, with a view to developing systems for different targeting and specific applications.

### 4.1. Polymeric Nanoparticles for MB

In the context of MB, polymeric NPs have been tested for both passive and active targeting. The first report on passive MB targeting concerned spherical poly(glycerol-adipate) (PGA) NPs fluorescently labeled with Rhodamine B Isothiocyanate (RBITC) [97]. It was shown that NPs were internalized by an endocytic process into the endosomal/lysosomal compartment in DAOY cells (SHH subgroup). After internalization, the probe was rapidly released, indicating breakage of the bond between the probe and PGA. It can be assumed that the lysosome acidic pH (∼4.5) was the cause of the breakage of the ester bond. Therefore, this study on NPs retention and metabolism indicates that to obtain a slower release, it is important to consider the compartment in which the NPs accumulate and to create more stable bonds to prevent premature drug release.

After this report, research on nanoformulations against MB began to increase, especially for drug delivery.

A polymeric NP (NVA622) formulation of curcumin was reported by Lim et al. [98] to induce apoptotic cell death on DAOY (SHH subgroup) and D283-MED (G3/G4 subgroup) MB cultures and G2/M cell cycle arrest. They showed that more than one signaling pathway was affected by curcumin, including insulin-like growth factor 1 (IGF1), signal transducer and activator of transcription 3 (STAT3), serine/threonine kinase (AKT), and Hedgehog (Hh). Other works have indicated that these pathways appear to be involved in MB development. In particular, IGF-1, IGF-2 ,and IGF-1R are related to MYC-amplified G3 MB [99], STAT3 is essential for Smo-dependent SHH signaling [100], and the PI3K/AKT pathway is common in MB, especially SHH-MB, affecting cell proliferation and chemoresistance [101]. However, the study conducted by Lim et al. [98] has some limitations since potential side-effects on non-neoplastic stem cells were not investigated.

In 2012, Chenna V. et al. [102] investigated the nanoencapsulation of HPI-1, an Hh pathway inhibitor, which has been demonstrated to block signaling downstream of Smo. HPI-1 was encapsulated into NPs made of PLGA conjugated with PEG. These authors demonstrated how the NP form improved systemic bioavailability compared to the free drug after both oral and parenteral administration in mice. Moreover, they demonstrated the ability of the nanoformulation to cross the BBB.

Madala H. et al. [103] developed NPs made of monomethoxy PEG and PLGA to deliver disulfiram (DSF) by passive targeting. They studied the effects on DAOY cells, as a representative model of MB, and IMR-90 stem cells, as an in vitro BBB model. Interestingly, they also investigated the cytotoxic mechanisms of DSF NPs transport into cells, finding that internalization was mediated by clathrin-coated vesicles, and that the accumulation process could be divided into two steps: first into lysosomes and subsequently into mitochondria. The data are consistent with previously reported mechanisms [97]. Moreover, in vivo analysis on intracranial MB xenografts showed the formulation was able to efficiently deliver DSF to the brain with an increased plasma half-life, while significant regression was observed compared to the unencapsulated DSF.

In 2020, the first attempt to deliver plasmid DNA encoding a suicide gene encapsulated in poly(beta-amino ester) (PBAE) NPs to treat MB was performed by Choi J. et al. [104]. They compared different polymers in different cell lines (D425 for G3 MB subgroup), finding that not all polymers delivered efficiently to all of them. The cell-type specificity could be due to the polymer end-groups, which lead to different uptake pathways. Herpes simplex virus type I thymidine kinase (HSVtk) was selected as a drug and resulted in the controlled apoptosis of transfected MB cells and a better overall survival rate in an in vivo MB model.

In 2021, Hwang D. et al. [105] studied a nanoformulation of poly(2-oxazoline) micelles for delivering vismodegib. The efficacy of vismodegib is reduced after administration because of its high affinity to serum proteins, resulting in a protein-bound form of the drug unable to penetrate the BBB. Vismodegib binds to Smoothened and downregulates hyperactivated Hh and has already been approved by the FDA for the treatment of basal cell carcinoma (BCC). Moreover, free vismodegib appeared to exert non-selective cytotoxicity. Vismodegib-loaded micelles could, therefore, reduce this intrinsic effect. The results confirmed decreased tumor growth in a genetically modified mouse model of MB SmoM2 compared to control. In 2022, the same group studied the same NPs used to encapsulate a CDK4/6 inhibitor, palbociclib [106]. This drug is known to block the cell cycle at the G1 phase; however, its brain penetration is limited, and high doses are toxic. Compared to the drug alone, encapsulated palbociclib improved the survival rate of G-Smo model mice. However, after a 24 h cell cycle, progression resumed, and the tumor began to grow again. Therefore, to enhance its efficacy, the mTORC1 inhibitor sapanisertib was added beneficially to the treatment, suggesting that NPs also represent powerful systems for combinatorial therapy for SHH MB.

Khang M. et al. [107] developed a nanoformulation with the aim of treating leptomeningeal metastases associated with MB. This study confirmed that the application of NPs is advantageous for both therapeutic purposes and detection of their precise location. The drug talazoparib, a poly(ADP-ribose) polymerase (PARP) inhibitor, was loaded into poly(lactic acid) with grafted hyperbranched polyglycerol (PLA-HPG) nanoparticles. The system not only improved drug retention in the cerebrospinal fluid but can also be functionalized with radio- or fluorescent-labels to assess drug distribution in mice.

The nanoparticles discussed so far rely on a passive targeting mechanism, whereas in 2009, Veiseh O. et al. [108] studied a non-viral vector (P-PEG-AF-CTX) made of polyethylenimine (PEI) polymer, functionalized with PEG, Alexa Fluor 647 (AF) as fluorochrome, and chlorotoxin (CTX), to specifically bind metalloproteinase-2 (MMP-2). MMP-2 is highly expressed in the lipid raft of cancer cells of neuroectodermal origin. This facilitates minimization of off-target effects on healthy cells

A more complex architecture was tested by Kokil G. et al. [109], who focused their research on cationic peptide asymmetric dendrimers with the aim of overcoming the limitations of commercially available high-generation cationic dendrimers. In particular, they used peptide dendrimer-lipid hybrid systems, in which the side arm of the polymeric peptide was conjugated with a lipid (cholic and decanoic acid), without masking the cationic charge, essential for effective electrostatic interaction with nucleic acids. This system appeared to self-assemble as a micelle, improving the protection of the nucleic acid from nuclease. Among the tested lipid arms, cholic acid-conjugated peptide dendrimers demonstrated improvement in nucleic acid delivery efficacy on DAOY cells compared to commercial standards.

Tjandra K. J. et al. [110] sought to overcome the limitations associated with the poor biodistribution and pharmacokinetics of many NP-based systems by creating artificial vesicles with a bilayer membrane, starting from amphiphilic block copolymers, called polymerosomes. These polymerosomes of different sizes and shapes were functionalized with the heptapeptide FSRPAFL, a selective target for DAOY cells. Moreover, these authors also investigated how ligand density affected uptake, identifying a threshold to the targeting potential. In the study, only the interaction with cells was analyzed. The results suggest that this system could improve bioavailability as it is able to evade immune cell recognition. Wang Q. et al. [111] focused on SHH and Group 3 MB, especially, cases with amplification of GLI1/2 and c-MYC genes. GLI transcription was modulated using JQ1, which is a small molecule bromodomain-containing protein 4 (BRD4) inhibitor, which usually promotes the progression and regulation of cell growth and transcription in neoplastic cell proliferation. It was shown that JQ1 inhibited GLI1 and c-MYC protein expression in DAOY and HD-MB03 cells. JQ1 was encapsulated into carboxyl poly(ethylene glycol)-block-poly(2-methyl-2-benzoxycarbonyl-propylene carbonate) (PEG-PBC) conjugated with COG-13. All the tests were also conducted on mice showing significant decrease.

More recently, Kumar V. et al. [112] suggested that chemoresistant MB could be treated by PEG NPs loaded with MDB5 and SF2523. MDB5 is a potent SHH inhibitor, which bind to the Smo protein, while SF2523 acts as a dual inhibitor of BRD4/PI3K. Specifically, on DAOY cells and HD-MB03 (G3 MB) cells, treatment with MDB5 and SF2523 arrested MB cells in the G0/G1 phase and promoted apoptosis. Moreover, these authors demonstrated that decoration of NPs with ApoE-targeting peptide COG-13 (targeting LDL receptor) improved the uptake of drugs in both cell lines and induced decreased tumor burden in orthotopic MB-bearing mice with very low systemic toxicity. In this case, the functionalization with LDL receptor binding ligands delivered the drugs more efficiently than non-targeted drug-loaded NPs.

Other biopolymers have been investigated for MB. In 2023, fucoidan-based nanocarriers were investigated as active targeting agents [113] in order to facilitate vismodegib transport into the brain tumor microenvironment. These nanocarriers were able to overcome the problems of passing through the BBB by fucoidan polysaccharide coating that targeted endothelial P-selectin and, consequently, induced caveolin-1-dependent transcytosis. Moreover, the delivery was more efficient in mice pretreated with irradiation. Globally, this work reported selective and active transport of NPs into brain MB, suggesting a strategy for the targeted intracranial delivery of drugs.

As summarized in Table 1, polymeric NPs have been tested as drug delivery systems in MB, especially by passive targeting. Strategies of active targeting termed biomimetic functionalization seek to promote biocompatibility of the vector and to improve delivery in cancer cells. The examined targeting ligands are not specific for MB, suggesting their applicability also to other cancer types.

### 4.2. Lipid Nanoparticles

In the context of MB, lipid NPs were assayed as drug delivery systems (Table 2). To the best of our knowledge, only non-active targeting strategies have been evaluated. Specifically, MacDonald et al. [114] encapsulated imipramine blue (IB), an NADPH oxidase inhibitor, into liposomes and tested them both in vitro and in vivo. The study reported that the decrease in NADPH oxidase 4 (NOX4) levels, significantly high in SHH MB, induced cell death by necrosis and promoted chemosensitivity. Effects were confirmed in an SmoA1 mouse, a commercially available animal model of MB.

Cationic lipid–polymer hybrid (LPH) NPs were assayed by Guo Y. et al. [115] to examine the prolonged delivery of Smo targeting siRNA in an MB mouse model. This hybrid was combined with MB-FUS, an acronym for microbubbles (MB) as contrast agents, activatable by low-intensity focused ultrasound (FUS). Cationic NPs protected RNA from degradation and influenced siRNA circulation time to increase cell uptake, while MB-FUS, on the other hand, appeared to maximize delivery by reducing vascular and interstitial barriers.

Globally, despite the recognized efficacy of lipid NPs as biocompatible and versatile nanocarriers, their application in MB appears to be in an exploratory phase (Table 2), with limited studies and no evaluation of a specific active targeting strategy to date.

### 4.3. Inorganic NPs for MB

In MB, inorganic NPs and their hybrid systems have been tested to improve radiosensitization [116,117,118], drug delivery [119], bioelectronic medicines [120], photothermal therapy [121], bioimaging [122], intracellular molecular detection [123,124], and to develop theranostic platforms [125] (Table 3).

With respect to radiosensitization, Kievit F. et al. [116] studied a nanoformulation able to increase the sensitivity of brain tumor cells to γ-irradiation radiotherapy (RT). In particular, they developed an NP able to deliver siRNA to knock down apurinic endonuclease 1 (Ape1) expression in order to sensitize UW228-1 cells (SHH group) to RT. Ape1 is an enzyme responsible for the process of base excision repair and is related to radiation resistance in cancer [126]. The system was made of superparamagnetic iron oxide coated with chitosan, PEG, and PEI to bind siRNA and protect it from degradation before being delivered to the target region. Similar results were obtained by Liu Z. et al. [117] who developed a system by changing the core of the NPs from iron oxide to gold. The encapsulation of siRNA (high specificity, minimum off-target effects) into NPs allows their use in clinical practice, and the reported data strongly suggest that NPs should be useful for avoiding siRNA degradation upon injection into the blood. More recently, nanodiamonds (NDs)—a class of carbon NPs—have been reported as promising tools in assisting RT, and their surface modification by hydrogenation or oxidation appears to enhance the γ-ray responsiveness of DAOY cells [118]. Moreover, Varzi V. et al. [118] demonstrated that the combined treatment of hydrogenated NDs and γ-rays induced increased cell killing through Bax-independent activation of Caspase-3.

Inorganic NPs have also been studied for their potential application in drug delivery. A study on drug delivery through magnetic NPs was performed by Engelhard H. et al. [119], focusing on magnetic Au–Fe alloy cores with streptavidin binding sites coated with biotinylated etoposide as the chemotherapeutic agent. The advantage of using magnetic NPs is their ability to be remotely guided using a rotating magnet over human-scale distances. Compared to etoposide alone, the nanoformulation was reported to reach target areas more rapidly, reducing drug dilution. Although active targeting has not been considered, the possibility of remotely controlling drug distribution represents a significant advantage in terms of clinical applicability and therapeutic efficacy. More recently, Jain A. et al. [120] focused on the putative role of NPs in regulation of the intracellular trafficking of chemotherapeutic drugs. Interestingly, they found that AuNPs could act as transducers that interact with cellular membranes when they are polarized by alternating current. In this way, the authors were able to overcome the problem of endosomal trapping, promoting escape from the endosomal compartments by using external electrical stimuli.

In the context of photothermal therapy for MB, in 2008, Bernardi et al. [121] developed gold-silica nanoshells to selectively target malignant brain tumor cells (i.e., HER2-overexpressing MB cell line DAOY) for photothermal ablation. To achieve this, nanoshells were tagged with antibody against HER2. The nanoshells generated heat after exposure to NIR light, causing cancer cell death. It was demonstrated that the system is non-toxic unless cells are exposed to NIR light. This strategy allows the specific targeting of tumor tissue without affecting surrounding normal tissues.

The first attempt to use NPs for intracellular molecular detection in MB was made by Dudu V. et al. [123]. They studied a method for using Qdots for binding the activated extracellular domain of EGFR and detecting the intracellular activated EGFR subpopulations in DAOY cells (SHH group). EGFR is overexpressed in various cancers [127], among them MB, and has a role in tumorigenesis and metastasis/invasion. The study showed that the use of Qdots enables the identification of biological markers specific to the tumor type, grade, and chemoresistance, confirming the predominant endosomal signaling. The same authors [124] extended the research by linking the Sendai virus to cationic liposomes to obtain cytosolic delivery of targeted Qdots. Again, Qdots were linked to a monoclonal biotinylated antibody in order to specifically recognize an intracellular epitope of the EGFR. The new system increased the specific intracellular labeling of EGFR by 50% by avoiding endosomal trapping. Moreover, virus-based liposomes allow protein localization via Transmission Electron Microscopy (TEM) analysis.

Significant research efforts have been directed toward IONPs and their hybrid derivatives, with particular emphasis on surface functionalization strategies for active targeting. In 2010, Knight et al. [122] demonstrated that an antibody fragment which recognizes JC human polyomavirus T-antigen attached to dextran-coated IONPs could target specific T-antigen-positive cancer cells, particularly, BSB8 mouse MB cells. These NPs could be used as superparamagnetic MRI contrast agents. However, no study on drug release was performed. In 2015, Li Y. et al. [125] studied transferrin-conjugated poly-ethylene glycol and allyl glycidyl ether (PEG-b-AGE)-coated IONPs to target TfR in D556 (G3 group) and DAOY cells. The aim of the work was to create an MRI contrast agent with minimal off-target background, thanks to the anti-biofouling properties of the polymer. This is important because in blood, serum proteins can form protein coronas on the surface of NPs, preventing their specific action. The selectivity was achieved through functionalization with transferrin, considering that the representative cells overexpress its receptor. Therefore, this system could have a dual role as a drug delivery agent and as a potential imaging agent for the quantification of targeted biomarkers. Two years later, the same group [128] used the same NPs to separate, via biomarker recognition and immuno-magnetic separation, circulating cancer cells in bio-fluid samples.

The shape-dependent internalization of IONPs has been widely analyzed. Thamizhchelvan A. et al. [129] suggested that in cancer cells, including MB cells, rod-shaped NPs are better internalized, primarily via clathrin/caveolae-mediated and phagocytosis mechanisms. Moreover, different shapes appear to be more or less functional in active targeting, as reported by Orza A. et al. [130] who investigated the synthesis of iron oxide nanorods (IONRs) for biomarker-targeted cell capture and separation. They used TfR-functionalized IONRs to capture D556 cancer cells and compared the results with spherical IONPs. The shape of the NPs influenced the results, suggesting a better response for rod-shaped NPs. This shape-dependent cell uptake is probably due to an intrinsic property of cells, which show a preference towards nanorod endocytosis mediated by clathrin and caveolae/lipid rafts compared to spheres [131]. Therefore, it is important to finely tune the characteristics of NPs in order to obtain the best performance. In 2019, another study on IONRs for the immunomagnetic capture of biofluidic biomarkers was performed by Xu Y. et al. [132], by functionalizing IONRs with transferrin to target TfR. The extension to previous works relates to the application: the authors suggest that the nanoformulation could be applied to microfluidic chip-based biomedical devices, since it responds to an alternating rotational magnetic field with rotational motions in the microfluidic system by promoting liquid mixing in the microfluidic chamber.

To the best of our knowledge, the first and only attempt at identifying a strategy for theranostic applications of NPs in MB was made by Forgham et al., in 2024 [133]. In this study, IONPs were used to deliver siRNA (therapy) and provide contrast for MRI (diagnosis). They developed a pH-responsive IONP coated with both perfluoropolyether and oligoethylene glycol acrylate. Perfluoropolyether enhanced serum stability and cellular uptake, while oligoethylene glycol acrylate increased water solubility and biocompatibility. Globally, the reported results highlight the possibility of obtaining effective delivery of siRNA and detection via MRI at the same time.

As summarized in Table 3, inorganic NPs and derived hybrid systems have been mostly investigated for SHH-MB and fast-growing G3 MB. Specifically, multifunctional hybrid systems with controllable activity seem to be promising for achieving selective, localized, and “on-demand” therapeutic interventions.

**Table 3 pharmaceutics-17-00736-t003:** Inorganic NPs and derived hybrid systems for MB.

Targeting Mechanism	Year	Inorganic NPs or Hybrid	Tested Models	Application/Results	Ref.
Passive	2015	Iron oxide core coated with chitosan, PEG, PEI, loaded with Ape1 siRNA (NP:siApe1)	In vitro UW228-1 cells (SHH)	Radiosensitization combined with gene therapy/ Efficient silencing of DNA repair systems and increased DNA damage after irradiation	[116]
	2017	Gold NPs coated with PEG chitosan, PEI loaded with Ape1 siRNA	In vitro UW228-1 cells (SHH)	Radiosensitization combined with gene therapy/ Efficient silencing of DNA repair systems and increased DNA damage after irradiation	[117]
	2020	AuFe core with streptavidin binding sites conjugated with biotinylated etoposide	In vitro D283 (G3/G4)	Drug delivery/ Remote control of drug distribution by rotating magnet	[119]
	2023	Nanodiamonds	In vitro DAOY cells (SHH)	Radiosensitization/ Induction of apoptosis	[118]
	2024	Spherical IONP compared to IONR	In vitro D556 (G3)	Imaging probes and drug delivery/ Better internalization of IONR via clathrin/caveolae and phagocytosis	[129]
	2024	Perfluoropolyether (PFPE) polymer-engineered IONPs complexed electrostatically with *Plk1* siRNA	In vitro D425 (G3)	Theranostic/ Improved delivering and protection of siRNA providing an agent contrast for MRI	[133]
	2024	AuNPs associated with alternating current	In vitro DAOY cells (SHH) D283 (G3)	Bioelectronic medicines/ Endosomal escape	[120]
Active	2008	Gold-silica nanoshells targeting HER2	In vitro DAOY cells (SHH)	Photothermal therapy/ Selective killing of cancer cells after exposure to NIR light	[121]
	2010	Dextran-coated iron oxide nanoparticles targeting JC virus T-antigen	In vitro BSB8 mouse MB	Putative MRI probe/ Cellular internalization	[122]
	2011	Qdots binding activated EGFR	In vitro DAOY cells (SHH)	Diagnosis/ Identify level of activated, intracellular EGFR populations	[123]
	2012	Sendai virus-based liposome: Qdots binding intracellular EGFR	In vitro DAOY cells (SHH)	Delivery and bioimaging/ intracellular labeling of EGFR bypassing endosomal entrapment	[124]
	2015	Tf-conjugated PEG-b-AGE-coated IONPs targeting TfR	In vitro DAOY cells (SHH) D556 (G3)	Delivery and bioimaging/ coating and stabilization of the IONPs with anti-biofouling effect	[125]
	2017	Tf-conjugated PEG-b-AGE-coated IONPs targeting TfR	In vitro D556 (G3)	Biomarker-targeted imaging for cell separation/ Detection of circulating tumor cells with IONPs and anti-biofouling effect	[128]
	2017	PEG-coated spherical IONP compared to PEG-coated IONR targeting TfR	In vitro D556 (G3)	Diagnosis, biomarker targeted imaging for cell separation/ Better detection and immunomagnetic cell separation by Tf-IONR	[130]
	2019	Tf-conjugated IONPs targeting TfR	In vitro D556 (G3)	Diagnosis/ Improved efficiency of TfR-positive cells compared with commonly used commercial magnetic separation agents	[132]

### 4.4. Biomimetic Nanoparticles

Chronologically, research on biomimetic nanoparticles for medulloblastoma (MB) is confined to the past eight years (Table 4). In 2017, Catanzaro G. et al. [134] developed a biomimetic cisplatin drug carrier. They managed to obtain redox-responsive bovine serum albumin (BSA) NPs, which, in the presence of glutathione (GSH), release cisplatin after disruption of the disulfide bond-mediated intra-particle cross-links. Comparing drug released in normal cells with respect to DAOY cells, it was interestingly reported that the cisplatin release was significantly higher in MB tumor cells than in normal ones. This system was effective in the specific targeting of tumor cells by exploiting their typically elevated intracellular GSH levels. Albumin modifications or albumin-based hybrid nanoparticles can function not only as drug delivery systems but may also exert intrinsic anticancer effects. An example is polynitroxylated albumin (PNA) nanoparticles, which act as caged nitric oxide carriers. In this context, Soltys B. et al. [135] developed a caged nitroxide NP made of polynitroxylated albumin (PNA) to target leptomeningeal dissemination and metastasis in an SHH mouse model. This is the first work which focuses directly on metastasis treatment instead of the main tumor. Interestingly, not all metastatic districts responded well, probably because of the different journey of the drug along the BBB. Unfortunately, the underlying molecular mechanisms have not been directly investigated; however, a vascular effect has been proposed.

Major efforts have been made with lipoprotein-based NPs. In 2018, Bell J. et al. [59] focused on HDL NPs in order to bind the HDL receptor, scavenger receptor type B-1 (SCARB1), and prevent cholesterol signaling, which is fundamental for tumor growth and maintenance in the SHH MB subgroup. It is known that cholesterol can covalently modify Hh ligands [136] and Smo [137]; therefore, by preventing binding with its receptor, the Hh pathway could be impaired. Moreover, since overexpression of SCARB1 is associated with shorter overall survival in MB patients, it is important to find a strategy to block its signaling pathway. Kim J. et al. [138], similarly to the previous work, used high-density lipoprotein-mimetic nanoparticles (eHNPs) to cross the BBB and deliver sonidegib (LDE-225), a Smo inhibitor, to cells. They applied dual targeting by using Apolipoprotein A1 to target SR-B1 receptors on brain endothelial cells, and stage-specific embryonic antigen-1 (SSEA-1+) or CD15 in order to directly target the stem cell population. Evaluation of the efficacy was also validated in cells pretreated with free anti-CD15 and in the presence of BLT-1, which is an inhibitor of the HDL receptor SR-B1 and controls the flow of cholesterol in and out of cells. These tests allowed analysis of the mechanisms behind the journey of the drug inside cells, finding that it is released inside the endosome and then undergoes a process of acidification. In vivo tests also showed decreased tumor growth.

More recently, Lico C. et al. [139] developed tomato bushy stunt virus (TBSV) NPs with a genetically engineered surface with peptides to target SHH MB. Among various NPs, those with specific outer surface peptides were chosen to test increase in uptake, and NPs were loaded with Doxorubicin (DOX). Cytotoxicity was assessed on SHH MB cells. CooP peptide-linked NPs produced over 90% of cell deaths and were able to specifically target MB in vivo. Two years later, in 2023, the same group tested CooP-NPs in vivo [140], showing that encapsulated DOX is able to inhibit progression of MB pre-neoplastic lesions with a five-fold lower dose with respect to non-encapsulated DOX.

Taken together, the reviewed literature suggests that biomimetic nanoparticles have opened promising avenues for the targeted treatment of MB, particularly in addressing tumor selectivity, cancer stem cell populations, and metastatic lesions. As reported in Table 4, some systems appear to be effective in vitro or in acute treatments (e.g., within hours), but there is a lack of long-term results, which should be further investigated, especially in the case of viral and lipoprotein-mimetic systems.

**Table 4 pharmaceutics-17-00736-t004:** Biomimetic NPs for MB.

Targeting Mechanism	Year	Biomimetic NPs	Tested Models	Application/Results	Ref.
Passive, (functionally- addressed toward cancers)	2017	Redox-responsive BSA-NPs loaded with cisplatin	In vitro DAOY (SHH)	Drug delivery and targeted release/ Cancer toxicity according to glutathione content	[134]
	2023	Polynitroxylated albumin (PNA)	In vivo transgenic SHH MB mice (MAP mice)	Prevention of metastasis	[135]
Active	2018	HDL-mimetic NPs targeting SCARB1	In vitro DAOY cells (SHH) D283 (G3)	Therapy/ Modulated cellular cholesterol, reduction in cell viability and depletion of cancer stem cell populations	[59]
	2020	HDL-mimetic NPs loaded with sonidegib, targeting endothelial SR-B1 and MB-stem cell CD15	In vitro DAOY (human) and PZp53 (mouse) MB (SHH) Ex vivo and in vivo transgenic SHH MB mice	Drug delivery and dual targeting/ Facilitated and targeted cellular uptake of drugs	[138]
	2021	Chimeric tomato bushy stunt virus NPs (surface genetically engineered with peptides to target brain cells) loaded with Doxorubicin	In vitro primary murine cell culture (SHH) In vivo transgenic SHH MB mice	Targeted drug delivery/ Reduced drug dose for cell death; reached the tumor in a specific manner.	[139]
	2023	Chimeric tomato bushy stunt virus NPs (surface genetically engineered with Coop peptides)	In vivo transgenic SHH MB mice	Targeted drug delivery/ Reduced drug dose for therapeutic effects.	[140]

### 4.5. Molecular Target of Nanomedicine for MB

The revised chronological overview of nanomedicine applied to MB suggests the need for improved consideration of the molecular target. As previously reported, the biomolecule or pathway that the NP system is designed to modulate or detect should be relevant to intracranial tumors in general. This is the case for NPs that alter BBB permeability or target endothelial cells [108,111,112,113], and all the NPs that seem to act on vascularity, as reported for redox-responsive PNA [135]. Furthermore, the use of Qdots to target [123] could have a broad application. Globally, all these strategies have been tested, especially in SHH MB to overcome the problem of BBB integrity associated with this subgroup. As has been reported, often the loaded drug is already used for chemotherapy, but its encapsulation into NPs offers the possibility of selective release, as for cisplatin-loaded BSA-NPs [134], DOX-loaded chimeric tomato bushy stunt virus NPs [139,140], IB-loaded liposomes [114], magnetic Au-Fe with biotinylated etoposide [119], and AuNPs under electric current [120].

With respect to the MB subgroups, the targeting and delivery of molecules via NPs appear to provide a more selective and localized effect. Specific studies toward SHH MB include those reporting HDL NPs targeting SCARB1 [59], eHNPs delivering the Smo inhibitor sonidegib [138], LPH delivering SMO-siRNA combined with MB-FUS [115], and chitosan, PEG, and PEI nanoformulations binding Ape1-siRNA and coating superparamagnetic IONPs [116] or AuNPs [117]. Studies on the G3 group include those investigating theranostic platforms [133] and TfR-targeting IONPs with anti-biofouling effects applied for circulating tumor cell separation [125,128,130,132].

Briefly, the main innovations relate to advanced strategies, like dual targeting, redox-triggered release, or electric field-enhanced endosomal escape. Data analysis reveals some gaps, like the lack of targeting addressed toward MB, and underexplored targets for metastatic complications.

## 5. Conclusions

As shown throughout this review, MB cells usually develop resistance, and classic treatments are not always effective. Nanomedicine offers a promising strategy to overcome these challenges through targeted drug delivery, the EPR effect, and minimizing systemic toxicity. In these terms, it is crucial to identify the molecular pathways that lead to resistance and target them with nanotechnology-based approaches. Utilizing traditional therapy combined with nanomedicine, it may be possible to target potential resistance patterns. A key focus in this field is the development and optimization of nanocarriers, NPs tailored to deliver therapeutic agents directly to tumor cells while minimizing damage to healthy tissue. These nanocarriers have the potential to enhance drug bioavailability, overcome the BBB, and reduce systemic side-effects. This aspect is essential, especially for SHH MB, where the intact BBB limits the entry of drugs into the brain at therapeutic concentrations. It is possible not only to deliver drugs, but also to add specific features, such as imaging contrast agents, to allow for simultaneous diagnosis, monitoring, and treatment of cancer. This is what can be defined by the term “theranostic”.

Currently, there are no active clinical trials on the employment of nanomedicine for the treatment of medulloblastoma. However, the future of MB treatment is moving towards more effective, highly targeted and less invasive therapies that will not only increase survival rates but also significantly improve patients’ quality of life. As previously indicated, the shift toward dual-targeting and stimuli-responsive mechanisms reflects the field’s growing maturity and sophistication.

Furthermore, innovations in early detection and personalized treatment strategies will enable clinicians to tailor therapies based on the unique molecular and genetic profiles of each patient’s tumor, maximizing treatment efficacy and minimizing resistance.

## Figures and Tables

**Figure 1 pharmaceutics-17-00736-f001:**
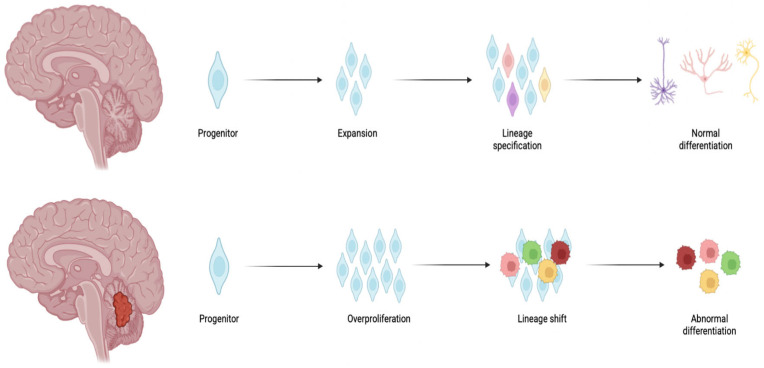
Graphical illustration of tumor development. Arrows illustrate the sequential flow of cell development in brain. Created with BioRender.com.

**Figure 2 pharmaceutics-17-00736-f002:**
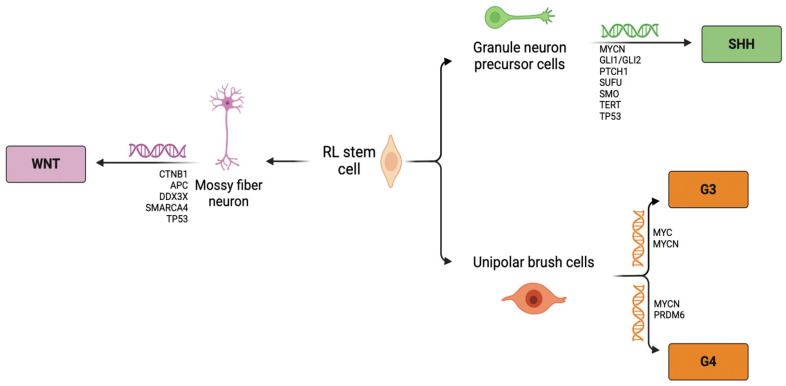
Schematic representation of MB subgroups classification. Arrows show molecular MB groups originated from RL stem cell (RL = Rhombic lip; WNT = Wingless; SHH = Sonic Hedgehog; G3 = Group 3; G4 = Group 4). Created with BioRender.com.

**Figure 3 pharmaceutics-17-00736-f003:**
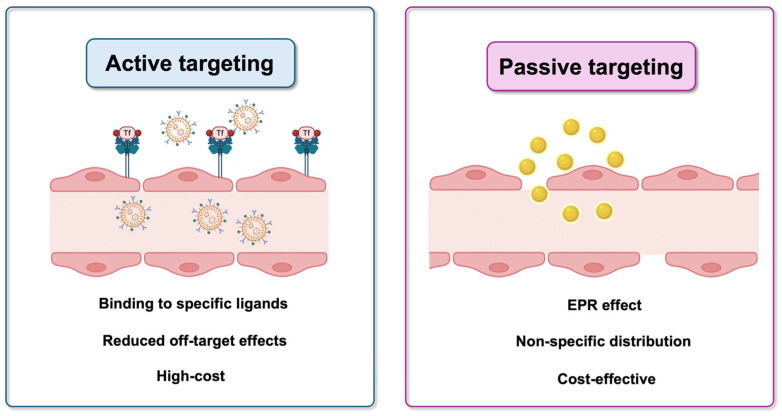
Representation of active and passive targeting mechanisms.

**Figure 4 pharmaceutics-17-00736-f004:**
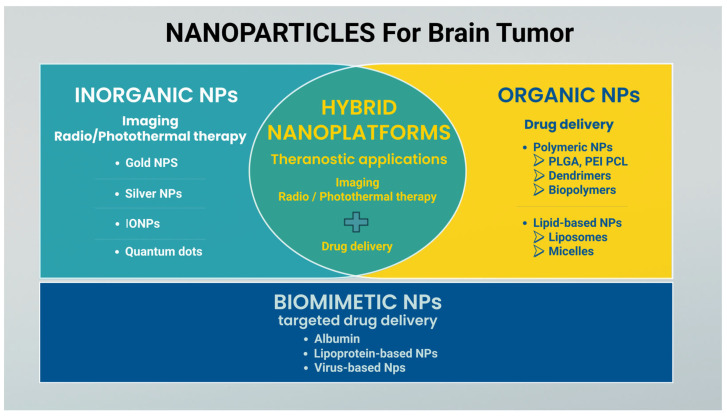
Classification of nanoparticles used for brain tumors. The main categories include inorganic, organic, hybrid nanoplatforms, and biomimetic NPs.

**Figure 5 pharmaceutics-17-00736-f005:**
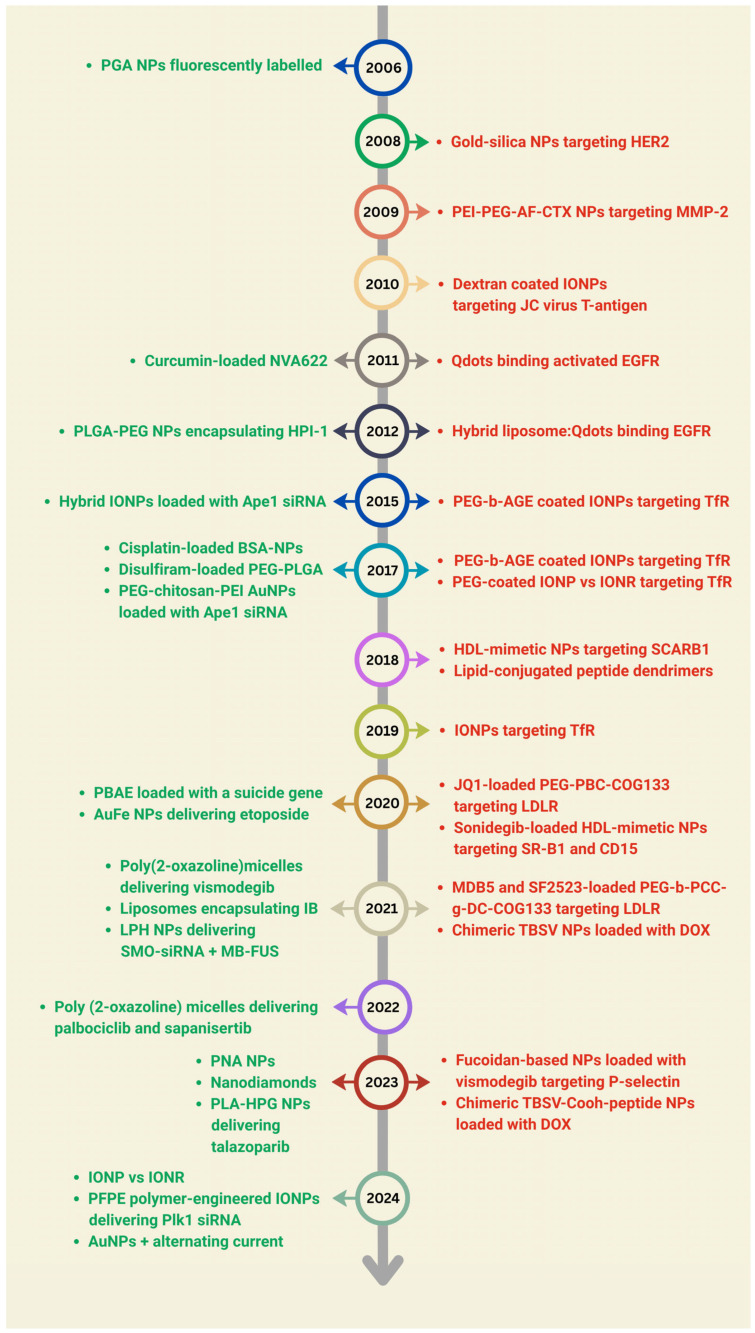
Timeline of NPs development for MB and their targeting mechanisms (from 2006 to 2024). NPs shown in green refer to passive targeting, while those in red refer to active mechanisms.

**Table 1 pharmaceutics-17-00736-t001:** Polymeric NPs for MB.

Targeting Mechanism	Year	Polymeric NPs	Tested Models	Application/Results	Ref.
Passive	2006	Spherical PGA labeled with Rhodamine B Isothiocyanate	In vitro DAOY cells (SHH)	Drug delivery/ NPs retention and metabolism	[97]
	2011	Synthetic polymer “NVA622” loaded with curcumin	In vitro DAOY (SHH), D283-MED (G3/G4)	Drug delivery/ Cell cycle arrest and apoptosis	[98]
	2012	PLGA conjugated with PEG encapsulating HPI-1	In vivo MB mouse model	Drug delivery/ Inhibition of allograft growth	[102]
	2017	Monomethoxy PEG and PLGA loaded with disulfiram	In vitro DAOY cells (SHH) In vivo MB mouse xenografts	Drug delivery/ BBB crossing and sustained drug supply by EPR	[103]
	2020	PBAE loaded with a suicide gene (herpes simplex virus type I thymidine kinase)	In vitro D425 cells (G3) In vivo MB mouse xenografts	Gene therapy/ Cell apoptosis and greater median overall survival in mice	[104]
	2021	poly(2-oxazoline) micelles to deliver vismodegib	In vivo transgenic mice that develop SHH-driven MB (G-Smo MB mice)	Drug delivery/ Extended overall survival and reduced systemic drug toxicity	[105]
	2022	poly(2-oxazoline) micelles to deliver palbociclib and sapanisertib	In vivo transgenic mice that develop SHH-driven MB (G-Smo MB mice)	Drug delivery and combinational therapy/ Extended mouse survival	[106]
	2023	PLA-HPG NPs dye-conjugated encapsulating talazoparib	In vivo xenograft MB mouse models	Drug delivery and bioimaging/ Metastasis treatment PET	[107]
Active	2009	PEI functionalized with PEG, Alexa Fluor 647 and chlorotoxin (CTX) to target MMP-2 (P-PEG-AF-CTX)	In vitro DAOY cells (SHH)	Gene delivery system for a broad array of cancer types	[108]
	2018	Lipid-conjugated peptide dendrimers (biopolymer, amino acids)	In vitro DAOY cells (SHH)	Delivery systems/ Effective improvement of the nucleic acid cargo and internalization	[109]
	2020	PEG-PBC loaded with JQ1 decorated with COG-133 peptide to target LDL receptor	In vitro DAOY (SHH), HD-MB03 (G3) In vivo orthotopic MB tumor in mice	Drug delivery/ Improved anticancer efficiency, inhibited MB progression	[111]
	2021	PEG-b-PCC-g-DC loaded with MDB5 and SF2523 (alone or in combination) decorated with COG-133 peptide to target LDL receptor	In vitro DAOY (SHH), HD-MB03 (G3) In vivo orthotopic MB tumor in mice	Drug delivery/ Hh pathway impairment and significant antitumor efficacy in chemoresistant MB	[112]
	2023	Fucoidan-based NPs (biopolymer polysaccharides) targeting endothelial P-selectin and encapsulating vismodegib	In vivo genetically engineered mouse SHH-MB model	Drug delivery and BBB targeting/ BBB crossing/ Hh pathway impairment/ Extended overall survival and reduced systemic drug toxicity	[113]

**Table 2 pharmaceutics-17-00736-t002:** Lipid and lipid-polymer NPs for MB.

Targeting Mechanism	Year	Lipid NPs	Tested Models	Application/Results	Ref.
Passive	2021	Liposome encapsulating Imipramine blue (Lipo-IB)	In vitro DAOY cells (SHH subgroup) In vivo SmoA1 transgenic mice (SHH)	Drug delivery/ Induction of cell necrosis and prolonged survival in mice.	[114]
	2021	Fluorescently labeled LPH delivering SMO-siRNA combined with MB-FUS	In vivo SmoA1 transgenic mice (SHH)	Gene therapy and combined strategy/ Enhanced siRNA delivery and efficiency (LPH) and increased BBB permeability (MB-FUS)	[115]

## Data Availability

Data sharing is not applicable.

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
