# Peer review of "Medulloblastoma: Molecular Targets and Innovative Theranostic Approaches"

_pharmaceutics, 2025, doi:10.3390/pharmaceutics17060736_

Round 1
Reviewer 1 Report
Comments and Suggestions for Authors
Scientific reviews are one of the most difficult types of articles to write. It is necessary to cover a wide range of articles, approaches and objects, while being focused on the target topic. Especially if such serious diseases in general and children's diseases in particular are affected.
The work has an extremely good background covering the last 20 years of studying the use of nanomedicine in the treatment and diagnosis of medulloblastoma.
The work is well structured, both systematically and historically.
The section Nanomedicine And Medulloblastoma: a chronological overview will be extremely interesting to a wide range of researchers.
Since the medical and biological direction is basically not my profile, I can say that part 2 of the review is easy to read and understand not only for specialists in the field of genetics.
As a specialist in the field of material science, I am most interested in the third section of the review. First of all, Au and Ag-based systems for phototherapy with subsequent PEGylation are considered. And standard approaches, such as encapsulation in liposomes, micelles and simple linear polymers.
In my opinion, it is necessary to take into account modern progress in the field of polymers of complex topology, such as dendrimers, Janus molecules, hyperbranched polymers, molecular brushes, etc.
Inorganic hybrids based on Fe, Co and Mn nanoparticles, which are widely used in MRI combined theranostics, are not considered.
Phototheranostic up conversion nanoparticles are not considered.
These comments only emphasize the huge amount of information that the authors still need to consider in the future.
The current one is obolozhenie and complete
The number of references to sources is more than 120, of which 20% relate to references 2020-2025. This would be a problem for standard reviews, but Marilena Brillat and Adrian Carol Eleonora Graziano's review includes some serious historical background.
Author Response
Response to Reviewer 1 Comments
|
||
1. Summary |
|
|
Thank you very much for taking the time to review this manuscript. Please find the detailed responses below. The corresponding revisions are written in red in the re-submitted files.
|
||
2. Questions for General Evaluation |
Reviewer’s Evaluation |
Response and Revisions |
Is the work a significant contribution to the field? |
|
Thank you. We really appreciate your feedback and we hope you enjoy the improvement we made based on your suggestions, as detailed in the point-by-point response. |
Is the work well organized and comprehensively described? |
|
|
Is the work scientifically sound and not misleading? |
|
|
Are there appropriate and adequate references to related and previous work? |
|
|
Is the English used correct and readable? |
The English is fine and does not require any improvement. |
|
3. Point-by-point response to Comments and Suggestions for Authors |
|
|
Comments 1: Scientific reviews are one of the most difficult types of articles to write. It is necessary to cover a wide range of articles, approaches and objects, while being focused on the target topic. Especially if such serious diseases in general and children's diseases in particular are affected. The work has an extremely good background covering the last 20 years of studying the use of nanomedicine in the treatment and diagnosis of medulloblastoma. The work is well structured, both systematically and historically. The section Nanomedicine And Medulloblastoma: a chronological overview will be extremely interesting to a wide range of researchers. |
||
Response 1: We sincerely thank the reviewer for this positive feedback. Comments 2: Since the medical and biological direction is basically not my profile, I can say that part 2 of the review is easy to read and understand not only for specialists in the field of genetics. Response 2: We thank the reviewer for this positive comment. Comments 3: As a specialist in the field of material science, I am most interested in the third section of the review. First of all, Au and Ag-based systems for phototherapy with subsequent PEGylation are considered. And standard approaches, such as encapsulation in liposomes, micelles and simple linear polymers. In my opinion, it is necessary to take into account modern progress in the field of polymers of complex topology, such as dendrimers, Janus molecules, hyperbranched polymers, molecular brushes, etc. Response 3: Thank you for pointing this out. We agree with this comment. Therefore, integrations have been made to include other types of nanomaterials, including those with complex topologies. We specifically included dendrimers and biopolymers (page 8; paragraphs 3.2.1, lines 290-299) that are relevant to give a context to better understand Section 4. Comments 4: Inorganic hybrids based on Fe, Co and Mn nanoparticles, which are widely used in MRI combined theranostics, are not considered. Phototheranostic up conversion nanoparticles are not considered. Response 4: Thank you for pointing this out. We partially agree with this comment. Therefore, we have added the concept of “hybrid systems” and their main applications (page 7 lines 264-266). hybrids based on Fe, Co, and Mn nanoparticles—commonly used in MRI-based theranostics—and phototheranostic upconversion nanoparticles were not included in section 3 of this review because they are not consistent with the scope and content of Section 4, which focuses specifically on nanomedicine strategies that have been investigated for medulloblastoma treatment. To our knowledge, these types of nanoparticles have not yet been substantially explored in the context of medulloblastoma, and including them would have introduced elements outside the targeted scope of this section. In order to avoid the misunderstanding of specialized readers like you are, we have added a clarifying sentence in the text to explain this choice, at the beginning of section 3: “A concise and critical overview of the most commonly used nanomedical models for brain tumors is essential in order to guide the readers toward the nanomedicine-based strategies that have been reported for MB” (page 5; lines 194-196). Comments 5: These comments only emphasize the huge amount of information that the authors still need to consider in the future. Response 5: We thank the reviewer for this valuable comments. The highlight on the vast amount of information that still needs to be considered is appreciated as input to guide future work. However, to reflect the latest developments in the field and provide a more comprehensive overview of the different materials, also paragraphs 3.2.3 has been uploaded (page 9-10), and it has been also added a general overview of Biomimetic NPs (paragraphs 3.2.4, lines 374-391). Overall, essential element for a better contextualization of scientific evidences related to the nanomedicine for MB were added. Comments 6: The current one is obolozhenie and complete. The number of references to sources is more than 120, of which 20% relate to references 2020-2025. This would be a problem for standard reviews, but Marilena Brillat and Adrian Carol Eleonora Graziano's review includes some serious historical background. Response 6: We appreciate the reviewer’s comment. The selection of references aims to capture the most recent advancements, while analyzing the substantial historical background. In Section 4, we decided to create paragraphs and new tables, in order to highlight the chronological overview related to each chemical group.
|
||
|
||
4. Response to Comments on the Quality of English Language |
||
Point 1: The English is fine and does not require any improvement |
||
Response 1: Thank for your feedback. |
||
5. Additional clarifications |
||
None |

Reviewer 2 Report
Comments and Suggestions for Authors
The review article focuses on the state of the art of innovative theragnostic nanomedicines for Medulloblastoma.
The authors should reorganize the sections, and the information provided in the article to make it more readable and understandable, as suggested below:
In section 2, dedicated to explaining the molecular medulloblastoma subgroups, the authors do not mention the MB tumour that overexpresses EGFR. Even if this kind of tumour does not appertain to a different classification group from the 4 explained, I recommend that it appears in this section as it is mentioned throughout examples of nanomedicines developed to treat MB in section 4.
In section 3.2. related to nanoparticle classification, several types of nanoparticles that are mentioned throughout section 4, are lacking. For instance, dendrimer-based NPs and polyethyleneimine NPs are not mentioned in section 3.2.1. The same applies to quantum dots that belong to the category of Inorganic NPs (3.2.3) and are not mentioned here. On the other hand, a new section for nanomedicines that are made from HDL particles or albumin should appear (because does not fit in any of the three categories already explained and it is an interesting approach since their use could diminish biocompatibility issues that arise with nanoparticles made of synthetic polymers and inorganic materials).
The section 4 despite the intention of the authors to show examples of nanomedicines for neuroblastoma developed chronologically, the information provided could be made more understandable by grouping these examples in sub-sections, according to the type of nanoparticle or if the targeting delivery of the nanoparticle given is active of passive. Another important thing to highlight is that in each sub-section, first show the examples of NPs tested in vitro and finally in xerographs models.
A new Figure that shows the key results on articles that had the best results in treating MB, mainly the one that belongs to the sub-groups with poor prognosis, should appear along with a figure scheme of the nanomedicine used.
Comments on the Quality of English LanguageThe quality of English could improve, certain grammar errors were found.
Author Response
Response to Reviewer 2 Comments
|
||
1. Summary |
|
|
Thank you very much for taking the time to review this manuscript. Please find the detailed responses below. The corresponding revisions are written in red in the re-submitted files.
|
||
2. Questions for General Evaluation |
Reviewer’s Evaluation |
Response and Revisions |
Is the work a significant contribution to the field? |
|
Thank you. We really appreciate your feedback and we hope you enjoy the improvement we made based on your suggestions, as detailed in the point-by-point response. |
Is the work well organized and comprehensively described? |
|
|
Is the work scientifically sound and not misleading? |
|
|
Are there appropriate and adequate references to related and previous work? |
|
|
Is the English used correct and readable? |
The English could be improved to more clearly express the research. |
|
3. Point-by-point response to Comments and Suggestions for Authors |
|
|
Comments 1: The review article focuses on the state of the art of innovative theragnostic nanomedicines for Medulloblastoma. The authors should reorganize the sections, and the information provided in the article to make it more readable and understandable, as suggested below: |
||
Response 1: We sincerely thank the reviewer for the time dedicated to our work. Your suggestions have been accepted and a general reorganization of the sections was made as pointed out and described below. Comments 2: In section 2, dedicated to explaining the molecular medulloblastoma subgroups, the authors do not mention the MB tumour that overexpresses EGFR. Even if this kind of tumour does not appertain to a different classification group from the 4 explained, I recommend that it appears in this section as it is mentioned throughout examples of nanomedicines developed to treat MB in section 4 Response 2: We thank the reviewer for this valuable suggestion. As recommended, we have specifically mentioned the EGFR-overexpressing medulloblastoma subtype in Section 2 to provide better context for its relevance in later discussions in Section 4. This addition has been made at line 136. Comments 3: In section 3.2. related to nanoparticle classification, several types of nanoparticles that are mentioned throughout section 4, are lacking. For instance, dendrimer-based NPs and polyethyleneimine NPs are not mentioned in section 3.2.1. The same applies to quantum dots that belong to the category of Inorganic NPs (3.2.3) and are not mentioned here. On the other hand, a new section for nanomedicines that are made from HDL particles or albumin should appear (because does not fit in any of the three categories already explained and it is an interesting approach since their use could diminish biocompatibility issues that arise with nanoparticles made of synthetic polymers and inorganic materials). Response 3: Thank you for pointing this out. We agree with this comment. Therefore, integrations have been made. According to your suggestion, we have revised Section 3.2 to include the full range of nanoparticle categories discussed in Section 4, ensuring consistency and improving alignment with the applications described specifically for medulloblastoma. Specifically, to reflect the latest developments in the field and provide a more comprehensive overview of the different materials, dendrimers and biopolymers (page 8-9; paragraphs 3.2.1, lines 301-316), quantum dots (page 10, paragraphs 3.2.3, line 346-351), and Biomimetic NPs (paragraphs 3.2.4, page 10, lines 362-367) have been added as essential elements for a better contextualization of scientific evidences related to the nanomedicine for MB. Comments 4: The section 4 despite the intention of the authors to show examples of nanomedicines for neuroblastoma developed chronologically, the information provided could be made more understandable by grouping these examples in sub-sections, according to the type of nanoparticle or if the targeting delivery of the nanoparticle given is active of passive. Another important thing to highlight is that in each sub-section, first show the examples of NPs tested in vitro and finally in xerographs models. Response 4: Thank you for pointing this out. We agree with this comment. Therefore, the information provided have been grouped in sub-sections, according to the type of nanoparticle. Moreover, specific tables have been made to highlight the targeting mechanisms, Comments 5: A new Figure that shows the key results on articles that had the best results in treating MB, mainly the one that belongs to the sub-groups with poor prognosis, should appear along with a figure scheme of the nanomedicine used. Response 5: Thank you. Accepted and done. Key results are summarized in tables 1, 2 3 and 4 added to section 4. Moreover, a new figure with a chronological indication of NPs with passive and active targeting strategies has been realized (Figure 5, page 11). Moreover, the main considerations about the olecular targets of nanomedice for MB have been synthetized in paragraph 4.5 (page 20).
|
||
|
||
4. Response to Comments on the Quality of English Language |
||
Point 1: The quality of English could improve, certain grammar errors were found. |
||
Response 1: Accepted and done. |
||
5. Additional clarifications |
||
None |

Reviewer 3 Report
Comments and Suggestions for Authors
In manuscript “Medulloblastoma: molecular targets and innovative theranostic approaches” by Alice Foti et al. described the possible theranostic nanoplatforms combining targeted drug delivery and simultaneous imaging, enlightening their potential as tools for personalized medicine and the possible theranostic nanoplatforms combining targeted drug delivery and simultaneous imaging are revised, enlightening their potential as tools for personalized medicine. Manuscript is interesting but however, there exist lots of questions and problems in the manuscript. The writing of the paper should meet the standard of the journal.
Specific comments:
- Authors provide a general description, but not focused on article topic, for ex., in Sections 3.1.1, 3.1.2 – it is general information, please more focused on Medulloblastoma.
- Section 3.2. Nanoparticle classification – not a complete classification and not specified on Medulloblastoma. Please correct it.
- In Section 4. Nanomedicine And Medulloblastoma: a chronological overview please add Fig for Readers.
- Should include more discussion on the difficulties and priorities of the current research field, as well as the practical applications that have been and will be realized in this field.
- It is better to summarize each of the molecular targets for potential application of nanomedicine to overcome conventional treatment limitations. In Abstract section Authors wrote about it, but information in the text is not structured. In Table 1 presented the Cell target not molecular targets.
- The review should be more on compilation of discussion and figures.
Author Response
Response to Reviewer 3 Comments
|
||
1. Summary |
|
|
Thank you very much for taking the time to review this manuscript. Please find the detailed responses below. The corresponding revisions are written in red in the re-submitted files.
|
||
2. Questions for General Evaluation |
Reviewer’s Evaluation |
Response and Revisions |
Is the work a significant contribution to the field? |
|
Thank you. We really appreciate your feedback and we hope you enjoy the improvement we made based on your suggestions, as detailed in the point-by-point response. |
Is the work well organized and comprehensively described? |
|
|
Is the work scientifically sound and not misleading? |
|
|
Are there appropriate and adequate references to related and previous work? |
|
|
Is the English used correct and readable? |
The English is fine and does not require any improvement. |
|
3. Point-by-point response to Comments and Suggestions for Authors |
|
|
Comments 1: In manuscript “Medulloblastoma: molecular targets and innovative theranostic approaches” by Alice Foti et al. described the possible theranostic nanoplatforms combining targeted drug delivery and simultaneous imaging, enlightening their potential as tools for personalized medicine and the possible theranostic nanoplatforms combining targeted drug delivery and simultaneous imaging are revised, enlightening their potential as tools for personalized medicine. Manuscript is interesting but however, there exist lots of questions and problems in the manuscript. The writing of the paper should meet the standard of the journal. |
||
Response 1: We thank the reviewer for this feedback. Comments 2: Specific comments: - Authors provide a general description, but not focused on article topic, for ex., in Sections 3.1.1, 3.1.2 – it is general information, please more focused on Medulloblastoma. Response 2: We thank the reviewer for the observation. The sentence meets with our intent and we appreciate it. Globally sections 3 intended to provide a brief overview of the main types of NPs studied for the application in the context of brain tumors. This section aims to give context to readers who may not be familiar with the broader landscape of brain tumors and with the relevance of nanomedicine within the common hallmarks of these diseases (like BBB permeability, targeting, etc). We intended this section as a mandatory and essential overview of of the most commonly used nanomedical models for brain tumors. In this rich landscape, we chose to focus on the main systems which are consistent with the scope and content of Section 4, where the full body of literature on nanomedicine approaches for MB is addressed. To improve clarity, we have revised the introductory section, specifically at page 2 lines 80-82, to better emphasize their contextual role and to more explicitly direct the reader toward the focused analysis in Section 4. In order to avoid the misunderstanding of specialized readers like you are, we have added a clarifying sentence at the beginning of section 3 to explain this choice,: “A concise and critical overview of the most commonly used nanomedical models for brain tumors is essential in order to guide the readers toward the nanomedicine-based strategies that have been reported for MB” (page 5, line190-192). Comments 3: Section 3.2. Nanoparticle classification – not a complete classification and not specified on Medulloblastoma. Please correct it. Response 3: We appreciate the reviewer’s comment regarding Section 3.2. As previously reported, section 3 was structured to provide a general overview of nanoparticle classifications relevant to brain tumors, with the goal of offering a foundational understanding for all readers. According to your suggestion, we have revised Section 3.2 to include the full range of nanoparticle categories discussed in Section 4, ensuring consistency and improving alignment with the applications described specifically for medulloblastoma. Specifically, to reflect the latest developments in the field and provide a more comprehensive overview of the different materials, dendrimers and biopolymers (page 8-9; paragraphs 3.2.1, lines 301-316), quantum dots (page 10, paragraphs 3.2.3, line 346-351), and Biomimetic NPs (paragraphs 3.2.4, page 10, lines 362-367) have been added as essential elements for a better contextualization of scientific evidences related to the nanomedicine for MB. Response 2 and 3: Globally, we agree with both these comments that inspired us to ameliorate the description of our work organization in the introduction. According to your comment, we added “Therefore, in this review, we first introduce the readers to the topic MB by focusing on histological and molecular features of MB subgroups. Then, we critically analyzed the major application of nanomedicine in context of brain tumors with a critical evaluation of the distinct advantages for biomedical applications linked to targeting strategies and/or to physicochemical properties of nanoparticles (NPs). These finding supported the last part of the work in which we chronologically analyzed the scientific literature by reviewing the nanomedical approaches for MB.” (page 1, lines 81-87) Comments 4: In Section 4. Nanomedicine And Medulloblastoma: a chronological overview please add Fig for Readers. Response 4: Thank you. Accepted and done. A new figure showing the timeline of studies on the use of nanoparticles in MB from 2006 to 2024 has been realized (Figure 5, page 11). Comments 5: Should include more discussion on the difficulties and priorities of the current research field, as well as the practical applications that have been and will be realized in this field. Response 5: We thank the reviewer for this valuable comments. Section 4 was rewritten according to your suggestion. Comments 6: It is better to summarize each of the molecular targets for potential application of nanomedicine to overcome conventional treatment limitations. In Abstract section Authors wrote about it, but information in the text is not structured. In Table 1 presented the Cell target not molecular targets. Response 6: We thank the reviewer for this valuable suggestion. To enhance clarity and immediate understanding, we chose to add a paragraph “4.5. Molecular target of nanomedice for MB” to summarize the molecular targets relevant to the application of nanomedicine. Moreover, Tables 1, 2, 3 and 4 includes the mechanism of targeting and the related-molecular targets exploited for selective delivery in medulloblastoma (especially for active targeting). Comments 7: - The review should be more on compilation of discussion and figures. Response 7: Accepted and done. Two new figures, more discussion, and four tables were added.
|
||
|
||
4. Response to Comments on the Quality of English Language |
||
Point 1: The English is fine and does not require any improvement |
||
Response 1: Thank for your feedback. |
||
5. Additional clarifications |
||
None |

Round 2
Reviewer 2 Report
Comments and Suggestions for Authors
The authors of the review paper could respond to all my comments in the first paper revision.
They better organized the different paper sections and prepared 3 new Figures, as well as tables in the sub-sections added aiming to sub-group nanomedicines according to the NP nature/type.
This new version can be accepted in the present form.
Reviewer 3 Report
Comments and Suggestions for Authors
The authors have made good improvements to the manuscript, which I think will be accepted for publication.